# Could Collected Chemical Parameters Be Utilized to Build Soft Sensors Capable of Predicting the Provenance, Vintages, and Price Points of New Zealand Pinot Noir Wines Simultaneously?

**DOI:** 10.3390/foods12020323

**Published:** 2023-01-09

**Authors:** Jingxian An, Rebecca C. Deed, Paul A. Kilmartin, Wei Yu

**Affiliations:** 1Chemical and Materials Engineering, Faculty of Engineering, University of Auckland, Auckland 1010, New Zealand; 2School of Chemical Sciences, Faculty of Science, University of Auckland, Auckland 1010, New Zealand

**Keywords:** decision tree, machine learning method, Naive Bayes, New Zealand Pinot noir wines, price points, provenances, regions of origin, soft sensors, vintages

## Abstract

Soft sensors work as predictive frameworks encapsulating a set of easy-to-collect input data and a machine learning method (ML) to predict highly related variables that are difficult to measure. The machine learning method could provide a prediction of complex unknown relations between the input data and desired output parameters. Recently, soft sensors have been applicable in predicting the prices and vintages of New Zealand Pinot noir wines based on chemical parameters. However, the previous sample size did not adequately represent the diversity of provenances, vintages, and price points across commercially available New Zealand Pinot noir wines. Consequently, a representative sample of 39 commercially available New Zealand Pinot noir wines from diverse provenances, vintages, and price points were selected. Literature has shown that wine phenolic compounds strongly correlated with wine provenances, vintages and price points, which could be used as input data for developing soft sensors. Due to the significance of these phenolic compounds, chemical parameters, including phenolic compounds and pH, were collected using UV-Vis visible spectrophotometry and a pH meter. The soft sensor utilising Naive Bayes (belongs to ML) was designed to predict Pinot noir wines’ provenances (regions of origin) based on six chemical parameters with the prediction accuracy of over 75%. Soft sensors based on decision trees (within ML) could predict Pinot noir wines’ vintages and price points with prediction accuracies of over 75% based on six chemical parameters. These predictions were based on the same collected six chemical parameters as aforementioned.

## 1. Introduction

Pinot noir is the most planted red grape variety in New Zealand and the second most planted variety after Sauvignon Blanc. Meanwhile, the export sales of New Zealand Pinot noir have increased significantly over the past five years, with 10,282 million L exported in the 12 months to June year-end 2020 (https://www.nzwine.com/en/winestyles/pinot-noir, accessed on 10 November 2022), which has attracted the attention of the New Zealand Government as well (https://bri.co.nz/current-research/#pn, accessed on 10 November 2022). Based on the importance of Pinot noir wines to New Zealand, it is necessary to establish and maintain a good reputation for New Zealand Pinot noir wines, which could benefit consumers and winemakers. For example, when consumers purchase wines, extrinsic cues such as brand name, region of origin, vintages and prices could drive their purchase decisions [1]. For instance, Chinese buyers in Hong Kong viewed American wines as dignified, Japanese wines as inventive, and Chinese wines as inexpensive [2]. Compared to high-involvement consumers, low-involvement consumers are more likely to prioritize price over wine quality when making purchase selections [3]. Occasionally, wine consumers may not be able to obtain these product extrinsic cues as the wine label is damaged, or lacking information, or some wine consumers cast doubt upon the trustworthiness of certain extrinsic cues provided by wine merchants, such as price point. In addition, the number of mislabeled wines is on the rise, posing a greater danger to food safety, such as harmful wine additives and contaminants [4]. Thus, protecting wine consumers from commercial fraud is essential. For instance, it is a typical fraudulent activity in the commercialization of Chinese rice wine from varied geographical origins as Shaoxing rice wine [5]. The sale of counterfeit wines could be detrimental to consumers’ interests and the reputation of authentic wine merchants. Therefore, wine classification by identifying regions, vintages and prices is crucial for safeguarding the high economic value of wine products, preventing illegal labelling, protecting against wine counterfeiting and characterize region protected designation of origin (PDO) [6,7].

To address these issues, soft sensors for predicting wines’ product extrinsic cues have been developed. The soft sensor works as a predictive framework which encapsulates a set of input data which provides predefined output data, utilizing a selected machine learning method (https://www.frontiersin.org/articles/10.3389/fbioe.2021.722202/full, accessed on 10 November 2022). Machine learning methods provide a prediction of complex unknown relations between the input data and desired output data without relying on a predetermined equation (https://au.mathworks.com/discovery/machine-learning.html, accessed on 10 November 2022). Soft sensors are built using machine learning methods, alongside easy-to-measure variables to predict highly related difficult-to-measure variables [8]. For instance, a soft sensor employing the machine learning method artificial neural network (ANN) was used to predict Pinot noir wines’ retail price based on viticultural data with an R2 value 0.80 [9]. The soft sensor utilising machine learning method least-square support vector machine (LS-SVM) successfully predicted 1-year-aged, 3-year-aged and 5-year-aged rice wines based on alcohol content, titratable acidity (TA) and pH, with R2 values of 0.91, 0.82 and 0.96, respectively [10]. A soft sensor using the machine learning method partial least squares-discriminant analysis (PLS-DA) was used to predict vintages of New Zealand Pinot noir in 2010 and 2015 with R2 values above 0.8; however, soft sensors using PLS-DA can only predict the vintages of New Zealand Pinot noir wines in 2014, 2016 and 2017 with an R2 value 0.632, an R2 value 0.626 and an R2 value 0.674, respectively [11]. The combination of nuclear magnetic resonance (NMR) spectroscopy and the machine learning method random forest was able to accurately characterize Pinot noir produced by Eger and Villany with 100% accuracy [12]. The machine learning method PCA, in conjunction with phenolic compounds, could be used to differentiate 221 samples of unreleased (2019 vintage) commercial Shiraz, Cabernet Sauvignon, and Merlot wines from 10 distinct Australian regions [13]. Researchers have tried many different chemical parameters and machine learning methods to predict the regions, vintages and prices. However, few researchers have attempted to construct soft sensors to forecast the regions of origin, vintages and price points of Pinot noir wines based on the same chemical parameters. To build soft sensors that can forecast the region of origins, vintages, and price points with acceptable prediction accuracy, it is necessary to understand the relationship between chemical compounds and regions of origin, vintages and price points of Pinot noir wines.

Phenolic compounds are important components in wines that influence sensory attributes. Based on their chemical structure, phenolics can be separated into flavonoids and non-flavonoids [14]. Anthocyanins, flavonols, and flavanols (including flavan-3-ols) are flavonoids, while hydroxybenzoic acid, hydroxycinnamic acid, stilbenes, and phenolic alcohols are non-flavonoids [14]. Quantitatively and functionally, phenolic compounds play a vital role in wines. For instance, phenolic compounds have a considerable impact on sensory attributes. For example, phenolic acids like coumaric, caffeic, gallic, and protocatechuic acids can cause astringency sensations [15]. Certain phenolic compounds in wines, including syringic acid, vanillic acid, and ferulic acid, have been reported as having mild sweetness [16]. Apart from their influence on sensory attributes, there are close correlations between phenolic compounds and product extrinsic cues, such as regions of origin, vintages, and price points.

### 1.1. Regions of Origin and Phenolic Compounds

Regions of origin can influence the carbon flow to specific branch pathways in the flavonoid metabolism of grape berries, resulting in changes in the phenolic profile of wines [14]. In light of this circumstance, phenolic compounds have been utilized to determine the geographical origin. For instance, monomeric phenols could be used to identify Chinese wines’ origin [17]. Flavonols were used to differentiate wines from France and Spain, whereas phenolic acids and flavan-3-ols were used to separate Cabernet Sauvignon, Cabernet Franc, Merlot, and Syrah grown in China [18].

### 1.2. Vintages and Phenolic Compounds

Vintage corresponds to the harvesting year of the grapes. The impact of vintage on phenolic compounds can be attributed to two factors: the effect of weather on the grapes’ growing season and the effect of storage and maturation of the wine. Due to the effect of weather on the vines’ growing season, t-resveratrol levels in Shiraz wines produced in 2003 were higher than those produced in 2004 [19]. In 2016 and other chilly, rainy years, petunidin 3-O-glucoside and OH-tyrosol were more prevalent in Malbec wines, but astilbin, malvidin-3-O-p-coumarylglucoside, and quercetin-3-glucoside were more prevalent in the 2017 and 2018 vintages [20]. In terms of ageing, wine phenolics change as the wine ages in the bottle. In Pinotage wines, for instance, caffeic acid concentrations remained steady during the ageing process, while malvidin-3-o-glucoside levels decreased [18].

### 1.3. Price Points and Phenolic Compounds

Several studies have uncovered associations between phenolics and wine price points. For instance, Kassara (2011) discovered a correlation between the maximum skin tannin concentration and an increase in projected wine prices [21]. The wines with the highest economic value exhibited higher concentrations of the phenolic parameters measured [22]. However, researchers such as Wu (2021) discovered no relationship between the amounts of particular phenolics such as catechin, quercetin, resveratrol and wines’ prices [23]. The association between New Zealand Pinot noir wines’ retail price and phenolic compounds is yet to be explored.

According to the relevant literature, there appear to be close correlations between phenolic compounds and wine regions of origin, vintages and price points. However, it is unclear as to whether phenolic compounds could be used to construct soft sensors capable of simultaneously predicting the provenance, vintage and price point of New Zealand Pinot noir wines. In addition, although some soft sensors have been developed to forecast the price points and vintages of New Zealand Pinot noir wines, the previous wine sample size did not provide a representative sample across different vintages, regions and price points in New Zealand. Therefore, in this paper, soft sensors using machine learning methods, including Naive Bayes and decision trees should be rebuilt based on diverse regions of origin, vintages and price points, capturing the diversity of New Zealand Pinot noir wines. In this study, 39 commercially available Pinot noir wines were purchased from New Zealand. The key chemical input data from these wines were collected in order to concurrently characterize the regions of origin, vintages, and price points of New Zealand Pinot noir wines, enabling complete information to be available to the wine-buying public. Additionally, reasons why prediction errors exist in soft sensors to predict regions of origin, vintages and price points were analyzed.

## 2. Materials and Methods

### 2.1. Pinot Noir Wines

Thirty-nine samples of commercially produced New Zealand Pinot noir wines from five regions (Central Otago, Marlborough, Martinborough, Nelson, and North Canterbury) in triplicate (3 different bottles of each sample, a total of 117 of bottles) were analyzed, with retail prices ranging from NZ$10 to NZ$80 and vintages from 2011 to 2020. The Pinot noir wines were grouped based on price point with categories including low-priced (<NZ $30), middle-priced (NZ $30–60 NZ), and high-priced wines (>NZ $60), and into two vintage ranges, namely old vintages (≤vintage 2016) and new vintages (>vintage 2016). To completely comprehend and understand the influence of regions of origin, vintages, and price points on New Zealand Pinot noir wines, a large sample size of 39 different wines was necessary to provide a representative sample of Pinot noir from diverse provenances, vintages, and price points. All Pinot noir wines were purchased online from New Zealand retail stores, including Countdown, Caro’s Wine, Glengarry Wines and Black Market Wine NZ.

### 2.2. Analytical Measurement

Table 1 lists the basic methods to measure chemical data, which was used as input data to predict the New Zealand Pinot noir wines’ regions of origin, vintages and price points. A Shimadzu 2550 spectrometer was used to measure the colour parameters, total phenolics, total flavanols, total flavan-3-ols, total anthocyanins, total tannins, and the chemical age of Pinot noir wines. An edge pH meter was used to measure wine pH. These chemical measurements were conducted in triplicate. For more detailed information about Pinot noir wines, chemical reagents and analytical measurements, please refer to the Appendix A for Chemical measurement procedure.

### 2.3. Estimation of Regions of Origin, Vintages and Price Points

Seven experts (six men and one woman, aged 24 to 40, who had lived in New Zealand for at least six years and held a WSET 3 certificate) were invited to evaluate regions of origin, vintages (old/new vintages) and price points (low/middle/high prices) of 78 bottles New Zealand Pinot noir wines. One of the seven experts was a winemaker, three were international wine traders, two were graduate students in Viticulture and Oenology, and one was a member of the wine evaluation teaching team. All had a more than the 5-year history of wine involvement. According to the definition of wine experts from wine specialists Parr et al., these panels were wine experts [29].

This ethics application was approved by the University of Auckland Human Participants Ethics Committee with Reference Number UAHPEC2696. Before the sensory evaluation, all experts were informed that the University of Auckland had approved the sensory evaluation, and they were required to sign consent forms before the sensory evaluation. After signing consent forms, they used two consecutive days to finish this sensory evaluation. First day, they were required to evaluate the product extrinsic cues of 39 samples of Pinot noir wines (labelled as No. 1–No. 39 by researchers), and second day, they were required to evaluate another 39 samples of Pinot noir wines (labelled as No. 40–No. 78 by researchers). These wine numbers are labelled by researchers and experts never have the chance to see Pinot noir wines’ bottles. During the sensory evaluation, wine samples (30 mL) were poured into ISO standard tasting glasses, which were randomly labelled with a two-digit code by researchers to eliminate bias. Each taster spent approximately ten minutes to taste a single glass of Pinot noir wine and took a twenty-minute break after every ten glasses of Pinot noir wine. During the sensory evaluation, experts used white papers against natural light. Soda water was provided to cleanse the palate, and coffee beans were provided to refresh the nose. For Pinot noir wines’ product extrinsic cues, please check Appendix A.

### 2.4. Machine Learning Methods

#### 2.4.1. Build Classification Models to Predict New Zealand Pinot Noir Wines’ Product Extrinsic Cue

Wineinformatics incorporates data science and wine-related datasets, including physicochemical laboratory data and wine reviews, to discover useful information for wine producers, distributors, and consumers [30]. Typically, Naive Bayes, k-nearest neighbour (KNN), decision tree, support vector machine (SVM) and Random Forrest are used in wineinformatics (Table 2) [12,30,31,32,33,34,35,36].

In this paper, classification models including Naive Bayes, k-nearest neighbour (KNN), decision tree and support vector machine (SVM) were used to build soft sensors with the help of software Matlab Matlab R2019b-academic use (University of Auckland, Auckland, New Zealand) to predict New Zealand Pinot noir wines’ regions of origin, vintages, and price points based on chemical data including A420 nm (absorbance at 420 nm for yellow colour), A520 nm (absorbance at 520 nm for red colour), A620 nm (absorbance at 620 nm for blue colour), total flavan-3-ols (within the flavanols), total flavanols (includes flavan-3-ols), total tannins, total anthocyanins, total phenolics, A280 nm HCl (absorbance at 280 nm is positively correlated with total phenolics), A520 nm HCl (absorbance at 520 nm is positively correlated with total red pigments), chemical age (chemical age (chemical age = A520 nm HCl/A280 nm HCl) is correlated with vintages), alcohol content (obtained from wine labels).

#### 2.4.2. Select Important Key Chemical Parameters to Influence Regions of Origin, Vintages (Old/New Vintages), and Price Points

There is a compelling case for employing machine learning approaches to choose key chemical factors that have the potential to have a significant impact on regions of origin, vintages, and price points. It could help winemakers comprehend how to improve the quality of their products.

Extra trees classifier, Gradient boosting classifier, Extreme gradient boost (XGB), and Random forest (RF) classifier (as shown in Table 3) are among the most well-known feature selection methods used to understand important chemical data [37]. In this paper, feature selection methods were used to understand which chemical parameters could characterize New Zealand Pinot noir wines’ regions of origin, vintages (old/new vintages), and price points by using in-house developed codes based on Python. In the in-house developed codes, the following open-sourced libraries: pandas, numpy, matplotilib, and scikit-learn were used [38,39]. During experiments, chemical data including A420 nm, A520 nm, A620 nm, total anthocyanins, total flavanols, total flavan-3ols, total tannins, total phenolics, A280 nm HCl, A520 nm HCl, chemical age, pH, and alcohol content (from wine labels) always worked as input data regardless of regions of origins (five regions), vintages (old/new vintages) or price points (low/middle/high price) work as output data, respectively.

## 3. Results

The results of the chemical measurements for 39 commercial New Zealand Pinot noir wines (a total of 117 bottles Pinot noir wines) are summarised in Appendix A. According to Appendix A, it could be inferred that the majority of New Zealand Pinot noir wines had total phenolics ranging from 2506–4660 mg/L, total anthocyanins ranged from 54–156 mg/L, total tannins ranged from 504–1368 mg/L, total flavanols ranged from 584–1400 mg/L, total flavan-3ols ranged from 324–686 mg/L, ethanol ranging from 12.8%–13.8% (*v*/*v*), and pH ranging from 3.53–3.79, which provide comprehensive chemical ranges representing the diversity of New Zealand Pinot noir wines. Total phenolics in wines can vary based on the grape cultivar, viticultural practices, skin maceration temperature, and grape pomace contact time [40]. For example, total phenolics could be increased by adding whole clusters and stems to Pinot noir wines during maceration and fermentation [41]. Compared to other wine grape varieties, the concentration of stable, non-acylated forms of anthocyanins in Pinot noir grapes is low [42]. Anthocyanins are the main pigments of young red wines, and their contents in the wines depend on the polyphenolic richness of grapes and their winemaking procedures [43]. Moreover, during storage, monomeric anthocyanins may undergo interactions with tannins to form pigmented tannins [44].

These 39 samples Pinot noir wines employed in this study could provide a more representative New Zealand Pinot noir wine commercial dataset as opposed to the dataset generated by the machine learning method synthetic minority over sampling technique (SMOTE), which used 18 samples of New Zealand Pinot noir wines from two vintages (2013 and 2016) [37]. Importantly, the New Zealand Pinot noir wines used in this study were easily available for purchase by the general public, as consumers did not have to spend a considerable number of money at the wine auctions to purchase Pinot noir wines that had been aged for a long time. With the use of this comprehensive dataset, soft sensors could predict the regions of origin, vintages and price points of New Zealand Pinot noir wines that are more readily available and representative of those on the shelf.

### 3.1. Building Soft Sensors to Predict New Zealand Pinot Noir Wines’ Extrinsic Cues Based on Same Collected Chemical Parameters

Seven wine experts were also asked to anticipate 78 bottles of Pinot noir wines’ (39 samples in duplicate) the regions of origin, vintages, and prices of Pinot noir wines. Nonetheless, even experienced consumers familiar with sensory attributes have difficulty estimating the regions of origin, vintages and price points based on Appendix A. As shown in Appendix A, it was particularly difficult for experienced customers to evaluate the price points of Pinot noir wines exceeding sixty NZ dollars. Therefore, soft sensors were developed to predict the regions of origin, vintages and price points of New Zealand Pinot noir wines based on chemical data. Phenolics are essential chemical compounds found in wines that can be used as quality and authenticity fingerprints for grape species, origin regions, and vintages [18]. Therefore, soft sensors using Naive Bayes, decision trees, KNN, support vector machine and random forest were developed to predict the region of origin, price, and vintage for 39 samples New Zealand Pinot noir wines, primarily based on chemical parameters related to phenolics such as A420 nm, A520 nm, A620 nm, total flavan-3-ols, total flavanols, total tannins, total phenolics, A280 nm HCl, A520 nm HCl. Soft sensors utilizing Naive Bayes could accurately predict the regions of origin, vintages and prices of New Zealand Pinot noir wines. In contrast, soft sensors utilizing decision trees could accurately predict the vintage ranges (older/newer vintages) and price points of New Zealand Pinot noir wines.

#### 3.1.1. Soft Sensors Predicting New Zealand Pinot Noir Wines’ Regions of Origin

##### Building Soft Sensors to Predict New Zealand Pinot Noir Wines’ Regions of Origin

A soft sensor utilizing Naive Bayes is an effective classification model for predicting the five regions of New Zealand Pinot noir wines, including Central Otago, Marlborough, Martinborough, Nelson and North Canterbury, based on the 13 chemical data including A420 nm, A520 nm, A620 nm, total flavan-3-ols, total flavanols, total tannins, total phenolics, total anthocyanins, A280 nm HCl, A520 nm HCl, chemical age and alcohol content (obtained from wine labels) of 39 samples New Zealand Pinot noir wines (every sample has three bottles, a total of 117 bottles Pinot noir wines).

According to Figure 1, the soft sensor accurately predicts the origin of New Zealand Pinot noir wines 88% of the time. Among the 39 samples of Pinot noir wines, there were eight samples (24 bottles) from Central Otago, ten samples (30 bottles) from Marlborough, seven samples (21 bottles) from Nelson, seven samples (21 samples) from Martinborough and seven samples (21 bottles) from North Canterbury. The soft sensor had a dismal 20% prediction error across five regions for identifying whether or not Central Otago produces New Zealand Pinot noir wines. Surprisingly, a soft sensor is more likely to misidentify Pinot noir wines from Marlborough as Pinot noir wines from Central Otago. Of the 117 bottles, four Pinot noir wines from Marlborough were incorrectly labelled as Pinot noir from Central Otago. In addition, when determining whether or not New Zealand Pinot noir wines are produced in North Canterbury, the soft sensor had a relatively poor prediction error of 17.4%. Similarly, soft sensors may misidentify three bottles of Pinot noir wine from Marlborough as Pinot noir wine from North Canterbury. Among the five regions that produce Pinot noir wines, Marlborough has the most area, so it is likely that Pinot noir wines from Marlborough will be evaluated similarly to Pinot noir wines from Central Otago or North Canterbury [45]. Extensive research has focused on predicting the origin of Pinot noir wines using soft sensors. Using random forest classification, 55 lipids identified by Ultra-performance liquid chromatography-time-of-flight tandem mass spectrometry (UPLC-TOF-MS) were used to predict the origin of commercial Pinot noir wines from Burgundy, California, Oregon, and New Zealand with a predict accuracy of 97.5% [46]. Similarly, NMR spectroscopy combined with the support vector machine and random forest could be used to clarify of Hungarian wines’ geographical origin [12]. However, it is impractical for New Zealand winemakers to invest in expensive analytical instruments like UPLC-TOF-MS and NMR.

##### Important Chemical Parameters to Predict New Zealand Pinot Noir Wines’ Regions of Origin

Feature selecting methods, including Random forest, Extra trees classifier, Gradient boosting classifier, and Extreme gradient boost, are utilized to define the association between 13 chemical data and the locations of origin of New Zealand Pinot noir wines in Figure 2. Even though some of the results in Figure 2a–d differ, it is still possible to conclude that total flavan-3-ols and A620 nm were the most important chemical parameters among 13 chemical data for characterizing the regions of origin of New Zealand Pinot noir wines except Figure 2a. Previously, Sun (2015) discovered that flavan-3-ol profiles might be utilized to distinguish the regional origin of red wines [47]. In addition, the blue component (A620 nm) is attributed to free anthocyanins in the chinonic form or interactions between tannins and anthocyanins, and total anthocyanins may also be altered by climate conditions specific to an area [24,48].

#### 3.1.2. Soft Sensors Predicting New Zealand Pinot Noir Wines’ Vintages

##### Building Soft Sensors to Predict New Zealand Pinot Noir Wines’ Vintages

Based on the 13 chemical characteristics of 39 samples New Zealand Pinot noir wines in triplicate (a total of 117 bottles), the soft sensor utilizing a decision tree is an effective classification model for predicting the new vintage (made after vintage 2016) and old vintage (made before and including vintage 2016) of New Zealand Pinot noir wines. Comparing 39 samples using the 13 different sets of chemical data of New Zealand Pinot noir wines to other machine learning methods SVM, decision tree, Naive Bayes, KNN, and random forest, the soft sensor using Naive Bayes had the highest prediction accuracy for predicting the vintages of New Zealand Pinot noir wines.

According to Figure 3a, soft sensors utilising a decision tree can distinguish between new and old vintages of Pinot noir wines more easily due to the soft sensor’s prediction accuracy of 89.74% and classification tree structure from machine learning method decision tree in this soft sensor has displayed in Appendix A. The soft sensor using Naive Bayes had difficulty distinguishing between Pinot noir wines with vintages 2014–2018. Out of 117 bottles of Pinot noir wines, eight bottles were mislabelled as vintage 2016, and six bottles were mislabelled as vintage 2017. Prior to this study, soft sensors based on Random Forest were used to classify vintage 2015 and vintage 2016 of American Pinot noir wines using NMR spectroscopic input data with classification accuracies of 97.4% and 100%, respectively. However, only the vintage 2015 and vintage 2016 were chosen for this study [49].

##### Important Chemical Parameters to Predict New Zealand Pinot Noir Wines’ Vintage Points

According to Figure 4a–d, total anthocyanins and alcohol content (obtained from wine label) appeared to be the most important chemical parameters defining the vintages (old/new vintages) of New Zealand Pinot noir wines except for chemical parameters characterized by Extra trees classifier in Figure 4a. Total anthocyanins are one of the most significant phenolic compounds responsible for the colour of red wines [50]. Younger wines often have higher concentrations of total anthocyanins than older wines because tannins or flavan-3-ols may react with total anthocyanins to form stable pigments, which could explain why total anthocyanins are the most relevant criterion for judging Pinot noir wines’ vintages [51,52,53]. Additionally, Pinot noir has a distinctive anthocyanin profile, as it lacks acylated anthocyanins and has a substantially larger amount of malvidin-3-glucoside relative to other red wines [41].

Surprisingly, alcohol content is also a significant chemical data for describing the vintage characteristics of New Zealand Pinot noir wines. In alcoholic beverages, yeast ferments hexose sugars (fructose, glucose) sourced primarily from grapes to produce ethanol [54]. During ageing, the alcohol level of Pinot noir wines may undergo interactions with carboxylic acid functional groups to form esters [55].

#### 3.1.3. Soft Sensors Predicting New Zealand Pinot Noir Wines’ Prices

##### Building Soft Sensors to Predict New Zealand Pinot Noir Wines’ Prices

The soft sensor using a decision tree was shown to be a reasonable classification model for predicting New Zealand Pinot noir wines’ high price (>60 NZ dollars), middle price (30–60 NZ dollars) and low price (<30 NZ dollars) based on 13 chemical data of 39 samples (total 117 bottles) New Zealand Pinot noir wines. The soft sensor using Naive Bayes had a better prediction accuracy in predicting New Zealand Pinot noir wines’ actual price compared to other machine learning methods, including SVM, decision tree, Naive Bayes and KNN based on the chemical data of 117 bottles of New Zealand Pinot noir wines.

Referring to the explanation from Figure 1, it could be inferred that, among 117 bottles Pinot noir wines, 15 bottles New Zealand Pinot noir wines with high prices, 60 bottles New Zealand Pinot noir wines with low prices and 42 bottles New Zealand Pinot noir wines with middle prices in Figure 5a. In Figure 5a, soft sensors using decision trees were able to forecast Pinot noir wines with low prices more correctly than those with middle and high prices, with a prediction accuracy of 88.9% and the classification tree structure in this soft sensor has displayed in Appendix A.

For the high price category, only ten bottles of Pinot noir wines were accurately predicted as having high price points. In contrast, three bottles of Pinot noir with high price points are categorized as having middle price points. According to a prior study, the majority of middle- and high-priced New Zealand Pinot noir wines contain similar chemical data, which may explain why soft sensors perform poorly when predicting middle- and high-priced New Zealand Pinot noir wines. In addition, it can be deduced that, out of 42 bottles of Pinot noir wine with middle price points, seven bottles were misjudged as having low or high price points by soft sensors. Similarly, among the sixty bottles of low-priced Pinot noir wines, two bottles were rated as Pinot noir wines with middle price points. According to the results, both experienced customers and soft sensors could not reliably predict the high price of Pinot noir wines. It may be because the cost of some Pinot noir wines is impacted not only by wine quality but also by extrinsic cues like wine reputation. For example, Benfratello (2009) found that reputation significantly influences Italian premium wines, namely Barolo and Barbaresco than wines’ tastes [56].

In Figure 5b, among 117 bottles of New Zealand Pinot noir wines, three bottles of Pinot noir wines with retail price from 5–15 NZ dollars, 33 bottles Pinot noir wines with a retail price from 15–25 NZ dollars, 33 bottles Pinot noir wines with retail price from 25–35 NZ dollars, 18 bottles Pinot noir wines with a retail price 35–45 NZ dollars, 12 bottles Pinot noir wines with retail price 45–55 NZ dollars, six bottles Pinot noir wines with retail price 55–65 NZ dollars, six bottles Pinot noir wines with retail price 65–75 NZ dollars, and six bottles Pinot noir wines with retail price 75–85 NZ dollars. Based on confuse matrix in Figure 5b, it could be inferred that six bottles Pinot noir wines with retail price 15–25 NZ dollars, five bottles Pinot noir wines with retail price 25–35 NZ dollars are wrongly predicted as high retail price compared to actual retail price. Meanwhile, only one bottle Pinot noir wine with retail price 35–45 NZ dollars is judged as retail price 15–25 NZ dollars, and one bottle Pinot noir wine with retail price 45–55 NZ dollars is misjudged as retail price 25–35 NZ dollars.

##### Important Chemical Parameters to Predict New Zealand Pinot Noir Wines’ Price Points

According to Figure 6a–d, pH is the most relevant element in determining the price points of Pinot noir wines, except for the chemical data characterized by Extreme gradient boost in Figure 6d. Malolactic fermentation (MLF) is a secondary bacterial fermentation used to raise the pH of most red wines. The MLF procedure is particularly desirable because to the function it plays in changing the wine’s quality [57]. MLF, for instance, can increase wine flavour and texture, improve microbiological stability, and reduce wine acidity [58]. Furthermore, in Figure 6a, chemical age is another important chemical parameter which is positively correlated with the price points of New Zealand Pinot noir wines. In Figure 6b, total anthocyanins are another important chemical parameter which is positively correlated with the New Zealand Pinot noir wines’ price points. Total anthocyanins and chemical age are both related with SO_2_ nonbleachable pigment, which has a positive association with bottle wines’ prices [13]. Moreover, in Figure 6c, A280 nm HCl is shown to be a further important chemical parameter followed by pH to characterize the price points of New Zealand Pinot noir wines. A280 nm HCl is positively correlated with total phenolic compounds, which was positively correlated with the wine bottle price and total phenolics [21].

### 3.2. Potential Prediction Error Analysis

According to Figure 1, Figure 3 and Figure 5, it can be extrapolated that there are still some prediction errors for soft sensors to estimate regions of origin, vintages, and price points, which may result from improper input data and chemical variances in output data.

#### 3.2.1. Improper Input Data

The input data of soft sensors to forecast regions of origin were decreased based on the relevance of chemical data ranked by feature selection methods such as Random forest, Extra trees classifier, gradient boosting classifier, and Extreme gradient boost, as shown in Figure 2a–d. According to Figure 7a, it can be concluded that when the top 12 chemical data ranked by feature selection technique Random forest and Extra trees classifier are used as input data, the prediction accuracy of soft sensors is higher than when 13 chemical data are used.

Combined with Figure 4a and Figure 7b it is possible that only total anthocyanins and chemical age, which are key chemical parameters to influence vintage points based on Section 3.2.2, work as input data, and that the prediction accuracy of soft sensors to predict vintage ranges (old/new vintages) has increased in comparison to 13 chemical data working as input data. It appears that when critical chemical properties function as input data, soft sensor accuracy could be improved. Similarly, from Figure 6b and Figure 7c it can be deduced that pH, total anthocyanins, and total flavanols work as input data, the soft sensors to predict price points more accurately than the entire 13 chemical data.

#### 3.2.2. Chemical Variances from Output Data

Chemical parameters derived from a total of 117 bottles (No. 1–No. 117) New Zealand Pinot noir wines were characterized by PCA. There were total 39 samples and every sample contained three bottles of Pinot noir wines. In the PCA score plot, the distance between the two bottles of Pinot Noir wine is greater, indicating that the wines’ chemical variances are greater. According to Figure 8, it could be inferred that there were some chemical variances in chemical data, even three bottles like No. 1, No.78 and No.79 New Zealand Pinot noir wines from the same sample but different bottles. In these soft sensors, chemical data were used as input data, which could be resulted in the prediction error in soft sensors to predict New Zealand Pinot noir wines’ region of origin, vintages and price points

Notes: No. 1, No. 78 and No. 79 Pinot noir wines are from the same sample; No.2, No. 77 and No. 80 Pinot noir wines are from the same sample No. X, No. 79 − X, and No. 79 + X are from the same sample. The red dot is the middle point of three bottles of New Zealand Pinot noir wines from the same sample. Different coloured circles represent Individual Pinot noir wines’ PCA scores. Lines have connected three bottles of New Zealand Pinot noir wines from the same sample.

There are typically chemical variances amongst natural goods that are difficult to manage. Grapes, maceration, fermentation, and storage conditions affect these chemical variances in Pinot noir wines. The composition of Pinot noir grapes is determined by climatic elements (climate and pedoclimate), soil chemical and physical qualities, biological factors, and the results of viticultural management [59]. To improve wine phenolics, winemakers have tried various maceration techniques, such as microwave maceration of Pinot noir grape must [60]. On one hand, oak barrels impart a number of oak-related chemicals, such as ellagitannins, furfural compounds, guaiacol, oak or whisky lactone, and eugenol, into wine. On the other hand, atmospheric oxygen infiltration through the oak barrel permits the gentle oxidation of specific components, resulting in colour changes and modifying of the wine’s taste qualities [61].

## 4. Discussion

This present study was conducted to predict New Zealand Pinot noir wines’ regions, vintages and prices by machine learning methods based on collected chemical parameters with acceptable prediction accuracy (Figure 1, Figure 3 and Figure 5). With the help of feature selection methods, it could be concluded that measured chemical parameters could contribute to New Zealand Pinot noir wines’ regions of origin, vintages and prices (Figure 2, Figure 4 and Figure 6). According to Appendix A, when soft sensors using decision trees to predict New Zealand Pinot noir wines’ vintage ranges (old/new vintages) or price points, among 13 chemical parameters, only total phenolics, total flavanols, total flavan-3ols, A420 nm, total anthocyanins, and alcohol content are necessary chemical parameters to work as input data. When total phenolics, total flavanols, total flavan-3ols, A420 nm, total anthocyanins, and alcohol content were necessary chemical parameters to work as input data, the soft sensors could predict region, vintage and price with prediction accuracy 76.92%, 93.16% and 85.47%. The reason that collected chemical parameters could be used as input data to predict New Zealand Pinot noir wines’ product extrinsic cues simultaneously with the acceptable prediction accuracy, is because many chemical parameters, especially phenolics, have a close relationship with Pinot noir wines’ regions of origin, vintages and price points. Another important reason is that regions of origin and vintages both have intensive relationships with wine prices. Region of origin is a significant factor in determining wine price because each terroir has its own unique features and serves as an indicator of its collective regional reputation, which could affect wine price [62]. Meanwhile, there are two sources of vintage’s impact on wine prices. In the case of fine wine, it has been shown that prices are proportional to the weather conditions that led to the production of the vintage’s wines [63]. The vine’s behaviour fluctuates from year to year, and what makes a vintage good or poor may be the result of varying climatic conditions [64]. The majority of price and quality variation between vintages may thus be attributed to the weather. Wines of older vintages would be aged for an extended period, which could increase wine prices [65].

Another important finding was that according to Figure 7, it could be concluded that the greater quantity of input data applied to soft sensors is not indicative of their high prediction accuracy. For example, according to Figure 2a,b, removing total tannins from input data could improve the accuracy of soft sensors predicting the provenance of New Zealand Pinot noir wines. Pinot noir grapes have fewer total tannins than other grape varieties [63]. In light of this, some winemakers would like to add food additive tannins prior to fermentation to compensate for the lack of sensory attributes caused by a lack of tannins [66]. This could explain why removing the interfering chemical parameters could improve the accuracy of soft sensors.

## 5. Conclusions

In this paper, a total of 117 bottles of Pinot noir wines (39 samples in triplicate) from different regions, vintages and prices were selected to build soft sensors to predict New Zealand Pinot noir wines’ regions of origin, vintages and price points. These 39 samples could represent New Zealand Pinot noir from diverse provenances, vintages, and price points. This study shows that soft sensors may accurately predict the regions of origin, vintages (old/new vintages), and price points of New Zealand Pinot noir wines with prediction accuracies of more than 80% simultaneously when the whole 13 chemical parameters worked as input data. In addition, the soft sensor with a decision tree (within ML) performed better than the previous soft sensor using PLS-DA (belongs to ML) to predict New Zealand Pinot noir wine with vintage 2014 in this paper’s Introduction. In the interim, no soft sensor for predicting the provenance of New Zealand Pinot noir wines based on previous literature. Meanwhile, when total phenolics, total flavanols, total flavan-3ols, A420 nm, total anthocyanins, and alcohol content (obtained from wine labels) worked as input data, soft sensors could predict the regions of origin, vintages, and price points with prediction accuracies of more than 75% simultaneously. Total flavan-3-ols could successfully characterise New Zealand Pinot noir wines’ regions of origin, total anthocyanins, chemical age, and alcohol content (obtained from wine label) could successfully characterise New Zealand Pinot noir wines’ vintages, and pH could characterise New Zealand Pinot noir wines’ price points. According to subsequent research, understanding the critical chemical characteristics could further improve the prediction accuracy of soft sensors by removing the interfering chemical parameters could improve the accuracy of soft sensors.

The current study is limited to the typically consumed Pinot noir wines (from the price point and commercial availability). It is unclear at the moment if the soft sensor will be applicable or not to the oldest vintage or expensive Pinot noir wine, which will require further work.

## Figures and Tables

**Figure 1 foods-12-00323-f001:**
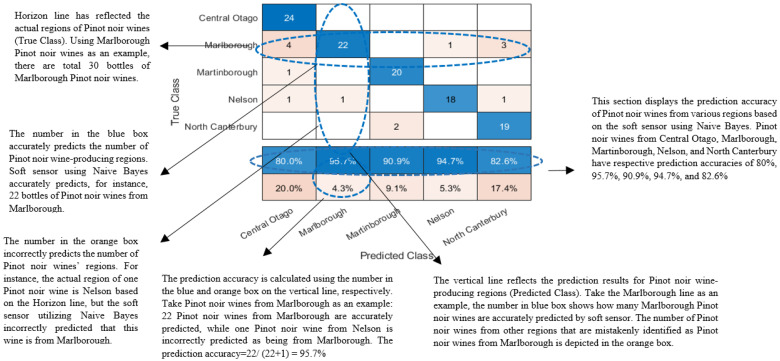
Confusion matrix about the soft sensor using Naive Bayes to predict 39 the region of origin of New Zealand Pinot noir wines. Dotted circles in the confusion matrix were used to help readers to understand what is True class, Predicted class, the meaning of the blue box and the orange box, and how the prediction accuracies are obtained.

**Figure 2 foods-12-00323-f002:**
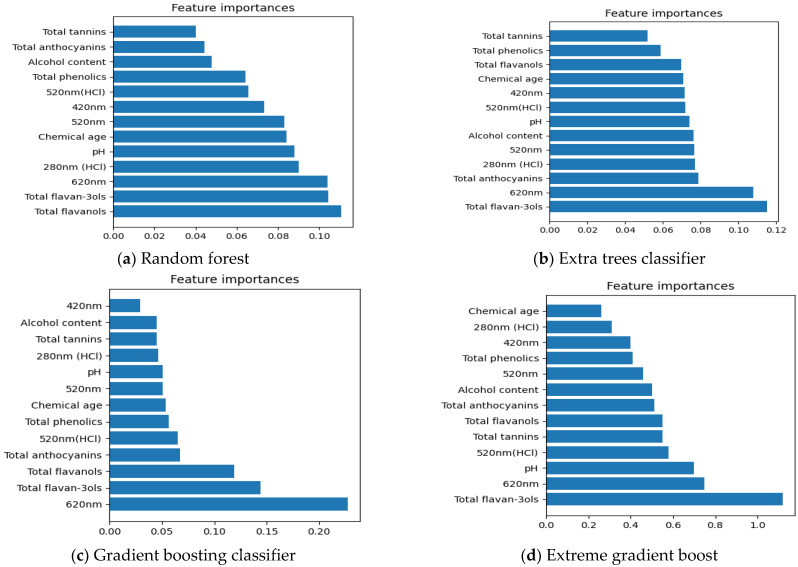
Features selection of 13 chemical data using Random Forest (RF), Extra trees classifier, Gradient boosting classifier, and Extreme gradient boost (XGBOOST) in regions of origin. (**a**–**d**) display the important chemical parameters in regions of origin that are selected by Random Forest, Extra trees classifier, Gradient boosting classifier and Extreme gradient boost, respectively.

**Figure 3 foods-12-00323-f003:**
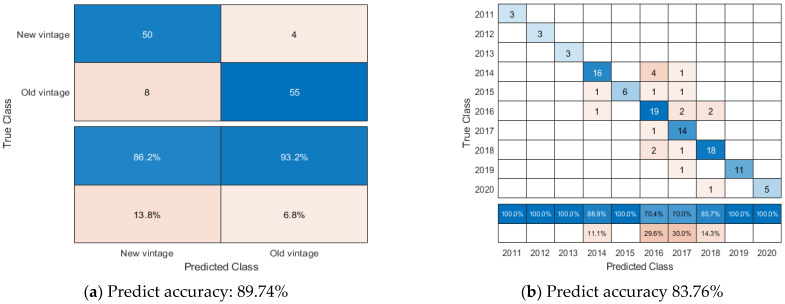
Confusion matrix about the soft sensor using a decision tree and Naive Bayes to predict New Zealand Pinot noir wines’ vintage ranges (old/new vintages) and vintages. (**a**) has displayed the confusion matrix of the soft sensor using decision tree when New Zealand Pinot noir wines’ vintages are classified into two categorize, namely old vintages (≤Vintage 2016), new vintages (>Vintage 2016). (**b**) has displayed the confusion matrix of soft sensors using Naive Bayes when New Zealand Pinot noir wines’ vintages.

**Figure 4 foods-12-00323-f004:**
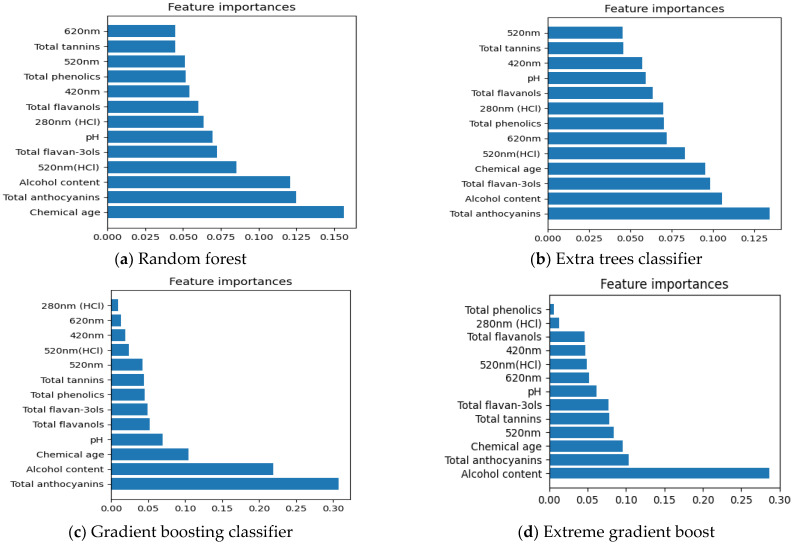
Features selection of 13 chemical data using Random forest, Extra trees classifier, Gradient boosting classifier, and Extreme gradient boost in vintages (old/new vintages). (**a**–**d**) display the important chemical parameters in Vintage Points that are selected by Random Forest, Extra trees classifier, Gradient boosting classifier and Extreme gradient boost, respectively.

**Figure 5 foods-12-00323-f005:**
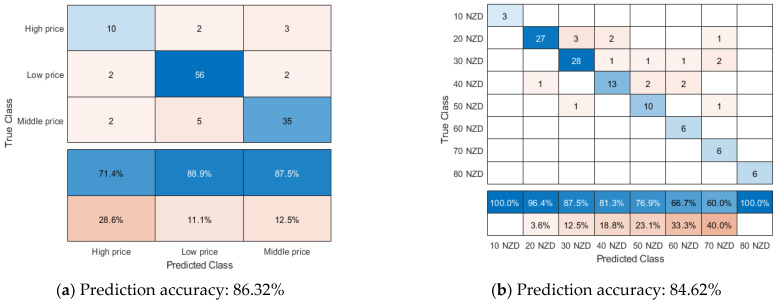
Confusion matrix about the soft sensors using a decision tree and Naive Bayes to predict New Zealand Pinot noir wines’ price points and actual price, respectively. (**a**) has displayed the confusion matrix of soft sensors using decision tree when New Zealand Pinot noir wines’ price are classified into three categorize, namely high price (>60 NZ dollars), middle price (30–60 NZ dollars) and low price (<30 NZ dollars). (**b**) has displayed the confusion matrix of soft sensors using Naive Bayes when New Zealand Pinot noir wines’ prices are classified into 10–80 NZ dollars. When retail price of New Zealand Pinot noir wine has ranged from 5–15 NZ dollars, this wine would be classified as 10 NZ dollars.

**Figure 6 foods-12-00323-f006:**
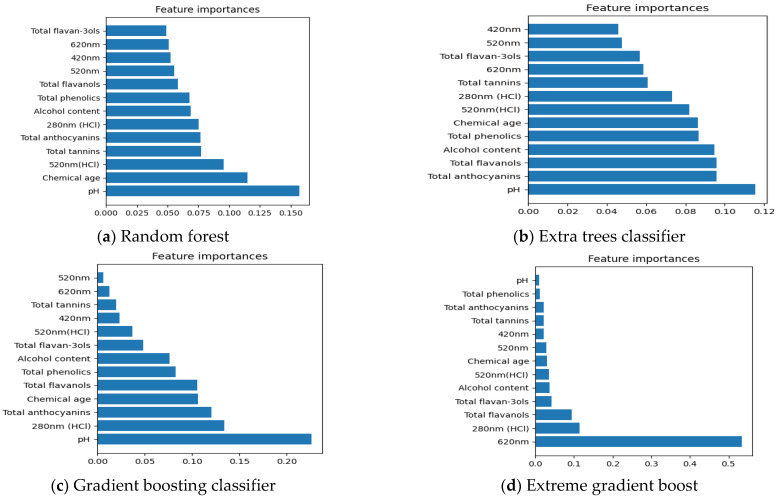
Features selection of 13 chemical data using Random forest, Extra trees classifier, gradient boosting classifier, and Extreme gradient boost in price points. (**a**–**d**) display the important chemical parameters in PricePoints are selected by Random Forest, Extra trees classifier, Gradient boosting classifier and Extreme gradient boost, respectively.

**Figure 7 foods-12-00323-f007:**
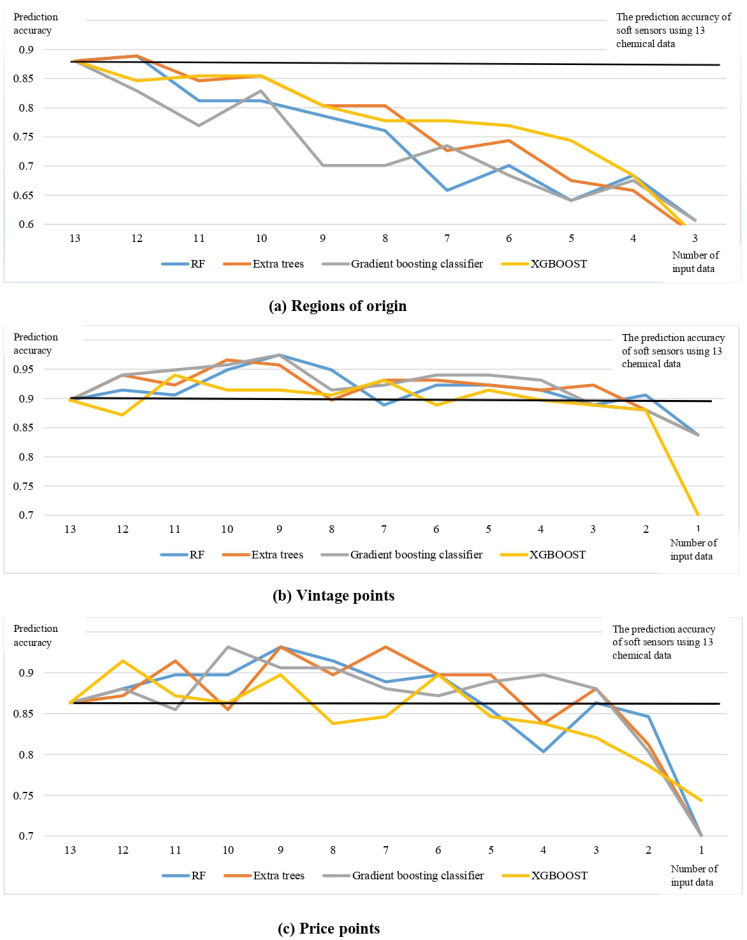
Prediction accuracy of soft sensors to predict regions of origin, vintage points, and price points with different input data. (**a**–**c**) display the prediction accuracies for soft sensors to predict New Zealand Pinot Noir wines’ regions of origin, vintage points and price points with different numbers of input data respectively. These input data were reduced with the help of the feature selection method Random Forest (RF), Extra trees classifier, Gradient boosting classifier and Extreme gradient boost (XGBOOST), respectively.

**Figure 8 foods-12-00323-f008:**
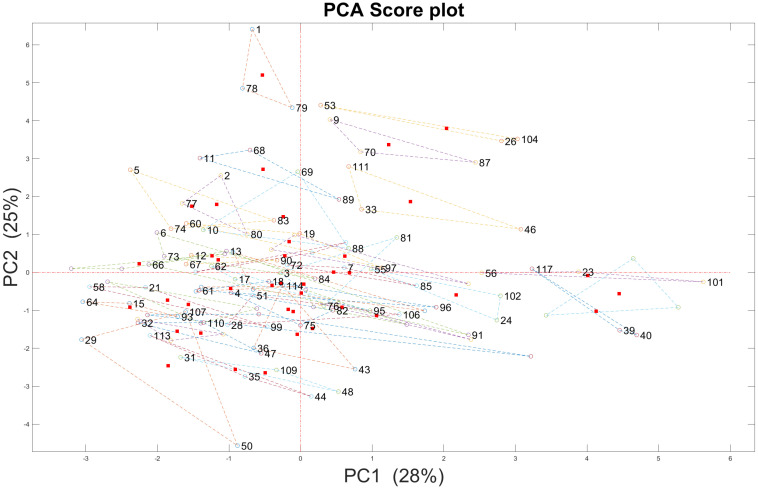
117 bottles New Zealand Pinot noir wines’ chemical parameters are characterized by PCA score plots.

**Table 1 foods-12-00323-t001:** Chemical measurements.

Experimental Aim: Data Collection	Reference
Colour measurement: A420 nm, A520 nm, A620 nm	[24]
Total phenolics assay	[25]
Total flavanols	[26]
Total flavan-3 ols	[27]
Total anthocyanins	[27]
Total tannins assay	[28]
Chemical age: A520 nm HCl, A280 nm HCl, chemical age	[24]
pH	

**Table 2 foods-12-00323-t002:** The usage of machine learning methods.

Classifier	Decision Tree	Naive Bayes	K-Nearest Neighbours (KNN)	Support Vector Machine (SVM)	Random Forrest (RF)
Definition	The decision tree tool is the most effective and widely used classification tool. A decision tree is a tree-like flowchart structure in which each internal node represents a test on an attribute, each branch represents a trial outcome, and each leaf node holds a class label.	Naive Bayes is a classification technique based on Bayes’ Theorem with the assumption of predictor independence [31]	K-nearest neighbour (KNN) is a method for classifying objects based on the training examples in the feature space that is closest to the target object.	Support vector machine (SVM) is an algorithm for classification and regression analysis in supervised machine learning.	Random forest is a supervised learning algorithm which creates decision trees on randomly selected data samples.
Traits	The benefit of the decision tree is that the mined information has high readability. Usually, important attributes are displayed at the top of the tree [30].	There is no relationship between the input data and attributes. Typically, input data or attributes influence the prediction of output data with equal weight [32]	K-nearest neighbour (KNN) is a simple, straightforward machine learning algorithm that can be used to solve classification and regression problems [30].	This classifier is a useful classification algorithm when there are few available training data and avoid overfitting [12].	Random Forest provides a fairly good indicator of feature importance.
Application	Wine grade [30]	Wine quality [33]	Regions of origin [30]	Wine quality [33] Region of origin [34]	Regions of origin [35,36]

**Table 3 foods-12-00323-t003:** The usage of machine learning methods to select important features.

Feature Selection	Extreme Gradient Boost (XGBoost)	Extra Tree Classifier	Gradient Boost Classifier	Random Forest
Definition	Extreme gradient boost is based on ‘boosting’, which combines all predictions of a set of ‘weak’ learners to develop a ‘strong’ learner through additive training strategies. In the meanwhile, XGBoost aims to prevent overfitting while optimizing computation resources by redefining the objective function and tree structure and optimizing the execution efficiency of the algorithm [7].	Extra trees classifier builds a set of unpruned decision trees using the standard top-down technique [37]	Gradient boosting builds new models from an ensemble of weak models, aiming to minimize the loss function [37]	See Table 2

## Data Availability

The data used to support the findings of this study can be made available by the corresponding author upon request.

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
