# Peer review of "Could Collected Chemical Parameters Be Utilized to Build Soft Sensors Capable of Predicting the Provenance, Vintages, and Price Points of New Zealand Pinot Noir Wines Simultaneously?"

_foods, 2023, doi:10.3390/foods12020323_

Round 1
Reviewer 1 Report
· For the most part, the abstract explains the purpose of the work and includes background information, but lacks a clear indication of the method used.
When describing the methods/algorithms used, it is recommended that the reader be introduced to the topic in a way that explains concepts such as soft sensors, Naive Bayes algorithms, etc. This is because many readers will not understand the purpose of the research (although it is very interesting and up-to-date).
It is also necessary to describe in a few simple steps the process of processing and obtaining results (especially chemical parameters) and the reason for choosing these parameters.
The sentences that introduce the reader to the topic of the research should be rephrased (lines 11-17). To do this, use the sentences in lines 110-115, which more clearly state the purpose of the research itself.
Line 18: Review the term "Nave Bayes."
· The introduction provides a good general background on the subject and gives the reader an idea of the wide range of possible applications of this technology. However, this chapter needs to be improved in the part where the soft sensors are described (lines 42-60).
· The methods used in this paper are appropriate for the purpose of the study.
Lines 144-145: Please provide a reference for this statement.
Line 153-156: When you list all the wine quality chemical parameters used in the analysis, you should group them and list them in parentheses. For example, if you mention color measurement (color determination at different wavelengths: A420nm for yellow color, absorbance A520nm for red color, and absorbance A620nm for blue color).
Lines 164-167: Please provide a reference for this statement.
The way the methods are currently written does not allow other researchers to replicate the experiment (because they are missing important details).
· Results: Many deficiencies were noted in the presentation of the results.
The results of the chemical measurements are not presented anywhere in the paper (not even in the supplementary material).
In the supplementary material, the first figure has no title.
Line 173: In addition, the authors mention sensory analysis. Maybe I missed it, but it is obvious that they are not included in this research.
Also, in the figures at a, b and c in the supplementary material, it is not clear what kind of results are involved and how they were obtained (lines 174-180).
Moreover, when the results are presented (e.g., Figure 4 a-d), it is necessary to accurately interpret the results shown there. In this particular case, the authors state the following "According to Figure 4 (a)~(d), total anthocyanins and chemical age are the most important chemical characteristics that define the vintage points of New Zealand Pinot noir 291 wines". However, if you look at Figure 4, this is not the case. The authors are asked to revise the written discussion and refer only to the results shown in their figures.
Many titles do not contain enough information to easily follow the text. Abbreviations should be avoided when naming figures, especially if they are not explained in the title. When looking at the results, the reader should follow the presented results independently, without paying attention to the abbreviations in the rest of the text.
· The conclusions in this paper need improvement. It is not clear how your research contributed to knowledge gaps, and there is no information about research limitations for future research.
In addition, there are minor differences in the presentation of the results obtained in the abstract and conclusion that need to be reconciled. For example, in the summary (line 20) the authors state "over 75%", while in the conclusion (line 460) they state "an accuracy of more than 80%".
Author Response
We appreciate the opportunity to revise and resubmit our manuscript titled ‘Could collected chemical parameters be utilized to build soft sensors capable of predicting the provenance, vintages, and price points of New Zealand Pinot noir wines simultaneously?’. We thank the editor and reviewers for the thoughtful comments and constructive suggestions, which we believe could improve the credibility of the revised manuscript. We have followed the editor’s suggestion and included page and line numbers to facilitate the reviewers in navigating through the responses. Accordingly, the revised manuscript has been systematically improved with new information and additional interpretations.
Please see the attachment

Reviewer 2 Report
In this paper, the authors considering soft sensors using Nave Bayes designed to predict New Zealand Pinot noir wines’ regions of origin, and soft sensors utilizing decision trees could predict New Zealand Pinot noir wines’ vintages, and price points both based on collected chemical parameters.
The focus of the study is interesting for winemakers, and wine consumers.
The paper is well written but some specific aspects should be reviewed.
In first, the authors indicate several times in the text that the analysis of wines including ‟total flavanols″ and ‟total flavan-3ols″. Why flavanols and flavan-3ols are different? I think that the authors should be more precise because both names are used for the same type of compounds.
In lines 61-62, the authors stated ‟Phenolics are one type of compounds, the third greatest component of wine, after water and alcohol″″. That is not true. Others compounds as glycerol or organic acids have higher concentration in the wine than phenolic compounds.
In lines 433-435, the authors noted ‟117 bottles of New Zealand Pinot noir wines’ 13 chemical parameter were characterized by PCA. Among 117 bottles (No.1~No.117), there were a total of 39 samples, and each sample contained three bottles of Pinot noir wines″. This sentence could be placed under Materials and Methods, because the initial reference to 39 samples and then 117 bottles is misleading.
Author Response

(The authors gave the same response as above.)

Reviewer 3 Report
In their Manuscript « foods-2076027» entitled "Could collected chemical parameters be utilized to build soft sensors capable of predicting the provenance, vintages, and price points of New Zealand Pinot noir wines simultaneously?" for Foods(journal), authors have reported somewhat original data and I think that the study is very interesting.
However, one mayor point and some minor points of criticism and questions have to be clarified:
MAYOR REVISION:
In my opinion, authors don’t give good reasons or present good objectives to do this work. I don’t arrive to understand at the end of my read why this work could be useful, even if I find it interesting. Lost wine labels or providing wine prices (lines 40-41) don’t seem me enough good reasons. Maybe mention some aspects about legislation/laws and origin of wines, or permitted grape varieties could help. But I think this point must be really reinforced.
MINOR REVISION:
Lines 22-28: too much long sentence, it should be divided into two sentences.
Lines 61: really is the third greatest component of wines? Always? Not acids or glycerol? Lines 77: I would replace this reference by other more recent.
Line 144 (and others…): The word “Naïve” appears not well written. This point should be corrected.
Lines 176-178: good comment! :)
Figure 1: “Predicted class” mention is not clear, it should be corrected. Line 261: Naïve (and not Nave): typing mistake.
Figure 5b: I think it would be better include in X axe the name of predicted class (and not the codes 10-80).
Lines 333 and 339: I don’t know where de “four” and “six” (bottles of) numbers come from. This point must be clarified.
Line 444: Typing mistake: …
Lines 415-418 and lines 427-430 should be included in Conclusions fourth section.

Author Response

(The authors gave the same response as above.)
